# Role of Hydrogen-Charging on Nucleation and Growth of Ductile Damage in Austenitic Stainless Steels

**DOI:** 10.3390/ma12091426

**Published:** 2019-05-01

**Authors:** Eric Maire, Stanislas Grabon, Jérôme Adrien, Pablo Lorenzino, Yuki Asanuma, Osamu Takakuwa, Hisao Matsunaga

**Affiliations:** 1Univ. Lyon, INSA Lyon, CNRS UMR5510, Laboratoire MATEIS, F-69621 Villeurbanne CEDEX, France; stanisla.grabon@insa-lyon.fr (S.G.); jerome.adrien@insa-lyon.fr (J.A.); pablo.lorenzino@insa-lyon.fr (P.L.); 2Japan Casting & Forging Corporation, 46-59, Sakinohama, Nakabaru, Tobata-ku, Kitakyushu 804-8555, Japan; asanuma.yuki.964@m.kyushu-u.ac.jp; 3Department of Mechanical Engineering, Kyushu University, 744 Motooka, Nishi-ku, Fukuoka 819-0395, Japan; takakuwa.osamu.995@m.kyushu-u.ac.jp (O.T.); matsunaga.hisao.964@m.kyushu-u.ac.jp (H.M.); 4Research Center for Hydrogen Industrial Use and Storage (HYDROGENIUS), Kyushu University, 744 Motooka, Nishi-ku, Fukuoka 819-0395, Japan; 5AIST-Kyushu University Hydrogen Materials Laboratory (HydroMate), 744 Motooka, Nishi-ku, Fukuoka 819-0395, Japan; 6International Institute for Carbon-Neutral Energy Research (I2CNER), Kyushu University, 744 Motooka, Nishi-ku, Fukuoka 819-0395, Japan

**Keywords:** X-ray tomography (X-ray CT), 3D image analysis, damage, hydrogen embrittlement, stainless steel

## Abstract

Hydrogen energy is a possible solution for storage in the future. The resistance of packaging materials such as stainless steels has to be guaranteed for a possible use of these materials as containers for highly pressurized hydrogen. The effect of hydrogen charging on the nucleation and growth of microdamage in two different austenitic stainless steels AISI316 and AISI316L was studied using in situ tensile tests in synchrotron X-ray tomography. Information about damage nucleation, void growth and void shape were obtained. AISI316 was found to be more sensitive to hydrogen compared to AISI316L in terms of ductility loss. It was measured that void nucleation and growth are not affected by hydrogen charging. The effect of hydrogen was however found to change the morphology of nucleated voids from spherical cavities to micro-cracks being oriented perpendicular to the tensile axis.

## 1. Introduction

Hydrogen energy is strongly expected as a secondary energy, which can be produced from various renewable energy sources and does not result in carbon dioxide (CO_2_) emissions when used as energy fuel in a fuel cell. Thus, hydrogen has a good potential for playing a role in the future development of our ever-growing society. Fuel cell vehicles (FCVs) have recently been commercialized in some countries, and constructions of hydrogen refueling stations have also been promoted. In such systems, various components (e.g., vessels, valves, regulators and metering devices) are exposed to high-pressure hydrogen gas environment. For a safe use of such components, it is necessary to properly understand the degradation of strength properties caused by the interaction of hydrogen with the microstructure, since hydrogen can easily penetrate into the material and causes “hydrogen embrittlement”, e.g., ductility loss in tensile test [1,2] and acceleration of fatigue crack growth [3,4,5,6,7,8] in a number of metallic materials. In addition, the degradation mechanism should also be clarified to review existing standards and regulations reasonably based on scientific grounds.

Austenitic stainless steels are successfully used for components installed in high-pressure hydrogen refueling stations and embarked in FCVs. When the stability of austenite phase is relatively low, plasticity-induced phase transformation from austenite to martensite under plastic deformation, i.e., α’ martensitic transformation triggers hydrogen-induced degradation of mechanical properties. The transformed martensite phase has the potential to be a crack initiation site [9] and can be a dominant factor of enhancement of crack propagation [10]. Even in austenitic stainless steels with high austenite stability, i.e., AISI316L, the presence of hydrogen also affects void nucleation and its coalescence behavior under intense plastic deformation [1,2]. Thus, fracture behavior of austenitic stainless steels with the presence of hydrogen depends on the phase stability. Although there exist several observations or analyses on fracture surface or a cross section of hydrogen-charged specimens after rupture, in situ analysis should be more effective so as to understand process and mechanism of the crack or void nucleation and/or coalescence behavior at various strain levels.

Analyzing damage experimentally has recently gained new interest thanks to the availability of 3D imaging techniques applicable for the observation of materials. X-Ray Computed Tomography (X-ray CT) is the most versatile of these new techniques [11,12], even at the nanoscale [13]. The results of X-ray CT can be analyzed quantitatively [14] and yield crucial information about damage evolution in ductile materials [15]. X-ray CT experiments have shown the crucial importance of hydrogen pre-existing pores in standard aluminum alloys on ductile fracture [16]. The technique has also been used to study damage process in a non charged standard AISI316L stainless steel in [17]. It has, however, never been used for steels charged with hydrogen. The goal of the present paper is then to use X-ray CT, able to quantify nucleation and growth of cavities in ductile materials, to assess the effect of hydrogen charging on these two mechanisms. For this, in situ tensile tests in X-ray CT were carried out on different steels with and without hydrogen charging. Both qualitative and quantitative results will be presented in the paper.

## 2. Materials And Methods

### 2.1. Materials

In this study, two types of austenitic stainless steels were investigated: AISI316 and AISI316L. The choice of these two metals containing different amounts of carbon was motivated by one main consideration. It is well known that carbon strongly influence the stability of austenite [1,2]. The motivation of this study was to perform in–situ observation of the fracture process of hydrogen–charged austenitic stainless steels with different phase stabilities. The AISI316L was provided by NSSC (Nippon Steel & Sumikin Stainless Steel Corporation, Tokyo, Japan), in the form of a plate 50×2500×6100 (mm) in dimensions. It was solution-treated at 1120 ${}^{\circ }$C for 4 min and then water–quenched. The AISI316 was provided by Yakin as a plate again (30× 2000 ×4000 mm in dimensions), solution-treated at 1120 ${}^{\circ }$C for 15 min and then also water–quenched. The composition of these two materials given by the provider is shown in Table 1 and their tensile properties (before hydrogen charging, provided by the manufacturer) are summarized in Table 2. The tensile properties after hydrogen charging were not measured with macroscopic tensile tests but will be analyzed later thanks to the in situ tensile tests. The AISI316L contains 0.04% of carbon, whereas the AISI316 has a carbon concentration about 0.18%. The samples were charged with hydrogen by being exposed to 100 MPa hydrogen gas at 270 ${}^{\circ }$C for 200 h. We know from previous experiments [18] that this results in a hydrogen content of 99.7 mass ppm with a uniform distribution over the cross section of the specimen. Hydrogen exists in both Face-Centered Cubic lattice sites and trap sites. Hydrogen outgassing can be negligible since hydrogen diffusivity in the austenite phase at room temperature is extremely low (10–16 m${}^{2}$/s).

The samples were subsequently stored in a freezer for a few months before the in situ tensile tests could be carried out in the synchrotron. This was necessary because synchrotron access is difficult to schedule precisely in advance. Electron backscatter diffraction (EBSD) maps of the two samples were acquired for grain size characterization. *Post mortem* Scanning Electron Microscope (SEM) observation of the fracture surface of the broken samples was also carried out.

In total, our experimental data base was then composed of four different types of samples: AISI316 and AISI316L hydrogen-charged and non-charged.

### 2.2. Methods

Before charging, smooth samples, with a useful part of 1 mm in diameter and 5 mm in length, were machined for each of the different materials. Each of the two heads of the samples were threaded (M3) which allowed for screwing additional T shaped tabs that were used to connect the sample’s head to the tensile grips. The specimen surface was polished with emery paper and then with a diamond paste to obtain a mirror–like surface finish. After the charging, samples were kept at −85 ${}^{\circ }$C. However, it is noted that, even if the samples were kept at ambient temperatures, the outgassing effect could be negligible since hydrogen diffusivity of these austenitic stainless steels are extremely low.

The in situ tensile experiments were conducted at the European Synchrotron Radiation Facility (ESRF), using the tomography setup available at the ID19 beamline [19]. The tensile rig used earlier introduced in [20] was especially designed for X ray tomography in situ experiments. The cross head speed was set to 1 μm/s. Each sample was screwed between two grips. The rig was placed on the rotation stage, between the X-ray source and the detector (the sample to detector distance was about 15 cm). The sample was rotated around the rotating stage axis while a high number (2000) of 2D X-ray absorption radiographs were recorded by the detector, the pixel size of which was set to 0.6 μm. A PCO DIMAX edge^®^ camera (Kelheim, Germany) was used to digitize the attenuation images. Each 2D X-ray radiograph required an exposure time of 0.01 s per frame. Because of the high attenuation of iron, the energy of the monochromatic beam was set to 50 keV. Using a Filtered Backprojection algorithm implemented at the ESRF (PyHST [21]), this series of radiographs were combined to reconstruct a 3D digital image where each voxel (volume element or 3D pixel) represents the X-ray absorption at that point. Because X-ray tomography is a non-destructive technique, many scans of the same sample could be acquired allowing us to observe damage evolution at various values of the applied strain in the different samples. During the scan acquisition, the displacement of the tensile machine was stopped to prevent blurring. This means that we operated in the so-called *interrupted* in situ mode. The force sensor of the tensile rig provided a measurement of *F*, the force applied to the sample at each time. The total true strain at each step was calculated from the reconstructed images, as explained later.

The raw volumes obtained from the reconstruction of the tomography scans subsequently needed to be processed in order to be used for the characterization of ductile damage. All the image processing and analysis steps were performed using ImageJ (Freeware 1.48q, Rayne Rasband, National Institutes of Health, USA) [22], a specific freeware available to perform image processing of 3D volumes. The images were first processed by removing the ring artifacts. The second processing consisted in median filtering the volumes (isotropic size of the filter = 2 voxels). This decreased the noise induced by the experimental method. The median-filtered volumes were thereafter binarized by simple thresholding to separate the material phase from the void phase. Each pore of the volume was then detected and labeled using a dedicated image processing procedure. During the last step of the process (labeling), the cavities having a volume smaller than 10 voxels (= 2.16 μm${}^{3}$), likely to be confused with noise, were rejected from the analysis.

The central area of each tensile specimen, where damage is mainly concentrated, was cropped from the initial image for damage quantification. The cropped volume has been chosen in the undeformed state to be a cuboid volume of (300)${}^{3}$ voxels i.e., (180 μm)${}^{3}$. It can be assumed (and has been verified for instance in [23]) that this central sub-region undergoes the highest stress triaxiality state and the highest strain during the tensile test. The size was chosen to be sufficiently large for the elementary volume to be representative but also sufficiently small for the strain and triaxiality to be spatially constant inside this sub-volume. During the tensile test, the selected initial volume plastically deforms. The shape of the cuboid volume has been chosen in the present study to change and become more elongated. The calculation of the amount of change to apply to the cube was based on the macroscopic plastic deformation of the sample.

## 3. Results

### 3.1. EBSD

Figure 1 shows EBSD maps of the two studied alloys shown here to highlight the grain structure of the material. These were obtained by using a Schottky type FE-SEM (JEOL JSM-7001F) at an acceleration voltage of 15 kV. The grains have a similar size (around 100 μm) and a similar amount of twins can be observed in both samples so the microstructure complies with our expectations for these very well known stainless steels. This value of the grain size is rather large compared to the sample diameter, but there are at least ten grains along the diameter so a total number of about 100 grains in a given section of the sample. This is a sufficient number of grains to insure that the mechanical behavior is not strongly influenced by plasticity gradient effects. The assessment of the texture would be interesting but is out of the scope of this paper.

### 3.2. Hardening Curves from In Situ X-Ray Computed Tomography

The sample was mounted vertically in the rig, more or less aligned with the tensile axis, which in turn is more or less parallel to the rotation axis of the rotation stage. This axis is denoted “*z*” in the following. As already performed in [15,24], from the outer shape of the sample measured using X-ray tomography after segmentation at each deformation step, we could measure the section *S* of the sample (*S* being perpendicular to *z*) as a function of the position of this section along *z*. From this list of sections S(z), we could determine the coordinates of the location (center of mass) of the minimal section of the sample $S_{min}$. Because we recorded *F* at all times, it was then possible to precisely calculate the true stress σ inside this minimal section using the expression:(1)$$\sigma =\frac{F}{S_{min}}.$$

Assuming no volume dilation of the sample due to damage, it was also possible to precisely calculate the true longitudinal strain, ε, in $S_{min}$ using the following standard expression:(2)$$\varepsilon =\ln(\frac{S_{min}^{0}}{S_{min}}),$$
where $S_{min}^{0}$ is the value of $S_{min}$ in the initial tensile state. Figure 2 shows the true stress—true strain curves (the hardening curves) recorded during our experiments for all the samples tested. It should in principle start at zero true plastic strain, but, in our case, true strain and true stress were measured at each stop during the interrupted test, with a first step at ε close to 0.2. We have no precise measurement of the yield stress of these samples during the in situ tensile test.

The hardening curves are close together with AISI316 hardening a bit more, probably due to its higher C content. We have gathered the ductility values measured from these in situ tensile tests referred to as $\epsilon^{f}$ in Table 3. We also calculated in this table the decrease in ductility induced by hydrogen charging DDH, calculated as:(3)$$DDH=\frac{\left(\varepsilon_{Non-charged}^{f}-\varepsilon_{Hydrogen-charged}^{f}\right)}{\left(\varepsilon_{Non-charged}^{f}\right)}\times 100.$$

DDH is clearly higher for the AISI316 than for the AISI316L.

### 3.3. Qualitative Damage Evolution

Figure 3 shows (as a selected representative example) a volume rendering of the evolution of the cavities in the non-charged AISI316L. It shows similar features compared to what was already observed in [17] on the same type of material. Voids, nucleate grow and coalesce during the severe plastic deformation of the sample, especially in the central region of the notch. This typical evolution is also observed in all the different tested samples and will be quantified further in a subsequent section.

By analyzing carefully volume series like the one shown in Figure 3 for every type of sample, qualitative differences were observed depending on the nature of the material and hydrogen charging. To highlight these differences, Figure 4 compares reconstructed slices of four typical samples in the last step before fracture. It is very clear from these images that the non-charged samples (left column) exhibit a much higher ductility, the section reduction and necking being much higher. Voids have then nucleated, grown and coalesced (see the big coalescence event observed for the non-charged AISI316 sample). This behavior is very typical of ductile metals. The right column shows the effect of hydrogen-charging on the final deformation stage. As already highlighted by Table 3, ductility was clearly reduced (necking is much less pronounced). Micro-cracks perpendicular to the tensile axis are observed in these reconstructions (as can be seen in Figure 5a) in the AISI316 sample but to a lesser extent in the AISI316L.

Figure 5 compares 3D renderings of the hydrogen-charged AISI316 and AISI316L in the final stage just before coalescence. The figure clearly show that the AISI316 exhibits many more microcracks than the AISI316L sample in which the voids are elongated along the tensile axis. In the AISI316, cracking from surface can be observed.

### 3.4. Quantitative Damage Evolution

#### Void Nucleation

Void nucleation was firstly quantified by calculating the number of cavities $n_{c}$ in every cropped volume. The mean cavities density N per cubic mm was calculated by dividing $n_{c}$ by the value of the analysed sub–volume. Figure 6 shows the evolution of the void density *N* as a function of the true strain for AISI316L and AISI316. Two samples were tested for the non-charged AISI316L; they are both plotted on the figure and they show a similar behavior. Nucleation being quite exponential with strain in steels (already observed in [15]), we have plotted the results in a logarithmic scale. Note first that the number of nucleated cavities is rather small in these samples. This is because of the homogeneous nature of these materials which present very few inclusions where damage can nucleate. It can be seen that, in the non-charged state, N increases very slowly in the AISI316 steel. We have noticed that, surprisingly, for this particular AISI316 sample, nucleation was anomalously small inside the neck and was mainly located at the periphery of the sample, where strain and triaxiality are not at their maximum. This is probably because when nucleation is very scarce, as is the case in these two materials, the random nature of the location of the nucleation site can lead to such surprising observations. We have then decided to reject this sample from the rest of the quantification. From the measurement of the three other materials, where nucleation was more substantial, it appears that hydrogen charging has only a weak effect on the nucleation kinetics, as can be measured by X-ray CT. The dotted curve included in the figure also shows the results that we previously obtained in [17] for a standard non-charged AISI316L sample (of different origin than the one used in this study). For these previous measurements, we were using synchrotron X-ray CT with a pixel size of 1.6 μm so detection capacity was smaller. This explains that the observed number of cavities was smaller in previous study. Despite these differences, we believe that it is comforting to see that the slope of the different curves are similar. The fact that *N* decreases at high strain for the hydrogen-charged AISI316 can be attributed to coalescence.

### 3.5. Void Growth

In the 3D images, each void is composed of a certain amount of voxels and this allowed us to measure the volume of each cavity $V_{i}$. From this value, we have then calculated the equivalent diameter of a sphere exhibiting the same volume:(4)$$D_{eq,i}={\left(6V_{i}/p\right)}^{1/3}.$$

It has been shown in previous studies that growth could be easily estimated by quantifying the average value of the largest cavities assumed to remain the same from one strain step to the next [23]. Here, we have chosen to work with the 20 largest cavities. Figure 7 shows the evolution of the average equivalent diameter of the 20 largest cavities in the cropped volume as a function of the true strain for the AISI316L non-charged, and for the two charged materials. For the sake of clarity, we have rationalized the values of $D_{eq}$ by dividing it by its value at 0 strain $D_{0}$. Previous measurement [17] is also shown as a dotted curve. In terms of growth, we clearly show again here that hydrogen charging has a weak effect of the growth rate of the cavities.

### 3.6. Aspect Ratio of the Cavities

The aspect ratio of the cavities has been calculated for the two smooth hydrogen charged samples (AISI316 and AISI316L) just before fracture (same as those shown in Figure 4). For calculating the aspect ratio of the cavities, we used the following simplified formula:(5)$$Aspect\;ratio=\frac{L_{x}+L_{y}}{2L_{z}},$$
where $L_{x}$, $L_{y}$ and $L_{z}$ are the largest dimensions of the cavity along the different directions of the reconstructed volume (remember that *z* is the tensile axis).

The aspect ratio is smaller than one for cracks and higher when the cavities are elongated along the tensile direction. Figure 8 compares the histogram of the values of the aspect ratio for the 20 largest cavities, for the two charged materials. This comparison is performed for a similar deformation step close to fracture. This histogram confirms the visual impression in Figure 5 i.e., the cavities have a crack–like shape in the case of the AISI316 sample and much less for the AISI316L.

## 4. Discussion

It is presumed that the variation in the aspect ratio of voids in hydrogen charged specimens is dominated by void growth in AISI316L and is dominated by the mixture of micro crack propagation and/or void growth in AISI316. As mentioned above, the aspect ratio of 0–0.5 corresponds to micro cracking and that of 0.5–1.0 corresponds to a mixture of micro cracking and/or void growth. For the samples charged with hydrogen, in AISI316, micro cracking dominated the fracture process. By contrast, in AISI316L, the fracture process was dominated by void growth. As a verification of these fracture characteristics, Figure 9 shows the fracture surface of hydrogen charged and non charged specimens as observed using SEM. In non charged specimens, the ordinary void nucleation, growth and subsequent coalescence are the main processes of the fracture in both AISI316 and 316L. As a result, cup and cone fracture occurred. In hydrogen charged AISI316, fracture surface was predominately covered by so called “quasi cleavage” (QC) with some small and elongated dimples. It is noted that QC corresponds to the cavities with the aspect ratio of 0–0.5 in Figure 8. By contrast, in hydrogen charged AISI316L, fracture surface is covered by smaller dimples compared to the non charged case, and QC is rarely observed.

The stability of austenite phase influences the susceptibility of the material to hydrogen-induced cracking. It is well known that nickel is a stabilizer of austenite phase [25], i.e., AISI316L with Ni content of 12.09% has higher stability than AISI316 with that of 10.23%. In AISI316 with lower stability, the austenite phase can easily transform to martensite phase. This transformation occurs above a certain intensity of plastic strain. It is possible that the phase transformation under plastic deformation facilitated the QC in AISI316. It was reported that acceleration of crack growth in austenitic stainless steel corresponds to regions where α’ martensite phase is present ahead of the crack tip under load [10]. In this region, hydrogen diffusivity becomes extremely higher compared to that of austenite phase (∼10–16 m${}^{2}$/s) [18]. The crack propagates into the transformed martensite phase or interface between austenite phase and α’ martensite phase. Koyama et al. investigated crystallographical characteristics of crack propagation in the α’ martensite phase and revealed that the crack preferentially propagates along the {100} [9]. Figure 10 shows a proposition for a schematic illustration of the fracture mechanisms. In hydrogen charged AISI316, firstly, the QC is likely to be generated at specimen surface, then to successively propagate concurrently with the small voids nucleation at the central part of the sample. Subsequently, final fracture probably occurs accompanied with elongated voids. It should be noted that, in austenitic stainless steels, hydrogen out-gassing can be assumed to be negligible because hydrogen diffusion in austenitic stainless steel is extremely low at room temperature [18].

Some of the authors of this paper proposed that a combination of slip localization due to the presence of hydrogen and the phase transformation in the vicinity of the crack tip causes a successive crack propagation [1]. On the other hand, in hydrogen-charged AISI316L, the QC was not generated since the material has a high austenite stability. Therefore, voids nucleated in the central part where stress triaxiality was high, i.e., necked region, with a similar mechanism to non-charged specimen. Hydrogen made the voids easier to coalesce by local shear stress owing to the slip localization, which resulted in void sheet formation [26]. The details in the tensile fracture mechanism of austenitic stainless steels charged with hydrogen have been comprehensively discussed in the literature [2]. The set of results presented above clearly visualize and verify the mechanism of the hydrogen-induced degradation.

## 5. Conclusions

In this paper, we have studied at the microscopic level the effect of charging AISI316 and AISI316L steels with hydrogen. We have used in situ X-ray computed tomography tests to analyze the fracture process of charged and uncharged samples. Our main findings are as follows:The ductility is reduced by Hydrogen charging, in a more important way for the AISI316 sample.In this material, cavities quickly transform into cracks perpendicular to the tensile axis leading to early fracture.By quantifying damage, we have also shown that both nucleation and growth, are not strongly affected by hydrogen charging. This means that the microscopic evolution of damage is not accelerated by the presence of hydrogen.The only noticeable difference, and the explanation for the reduction in ductility, is the aspect ratio of the cavities showing again a crack shape in the hydrogen-charged AISI316 leading to earlier macroscopic fracture.

Given these conclusions and regarding potential applications, 316L is better for use in vehicle tanks.

## Figures and Tables

**Figure 1 materials-12-01426-f001:**
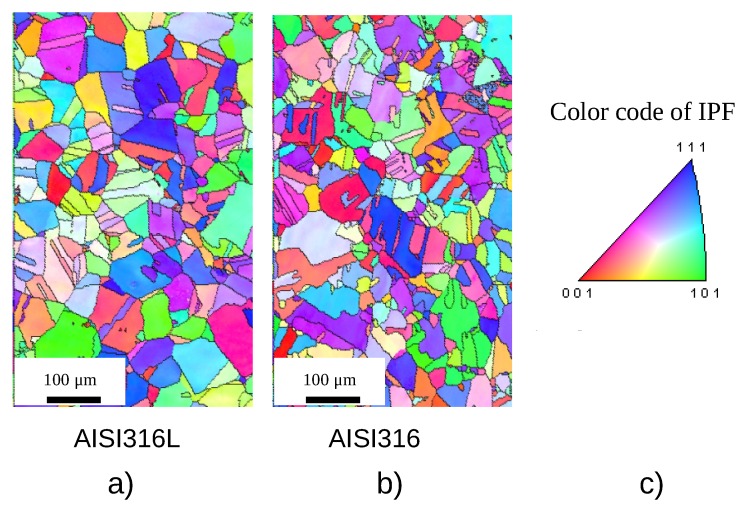
EBSD maps of the two different materials (**a**) non-charged AISI316L steel; (**b**) non-charged AISI316 steel; (**c**) inverse pole figure (IPF) coloring.

**Figure 2 materials-12-01426-f002:**
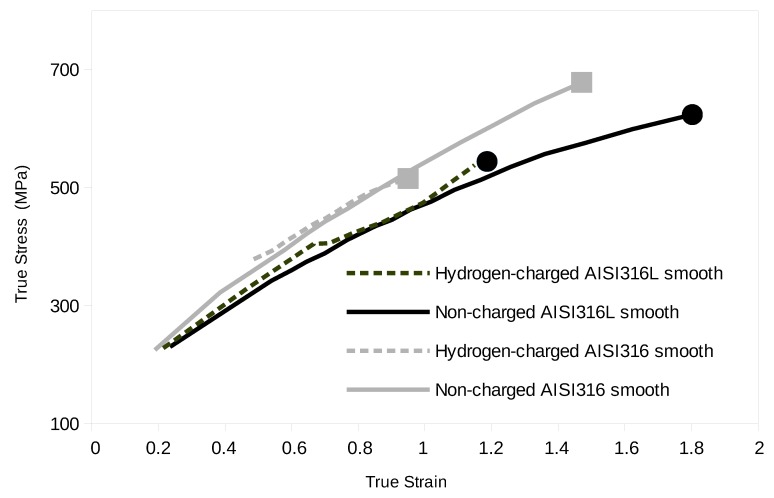
Tensile curve of the different samples (true stress vs. true strain).

**Figure 3 materials-12-01426-f003:**
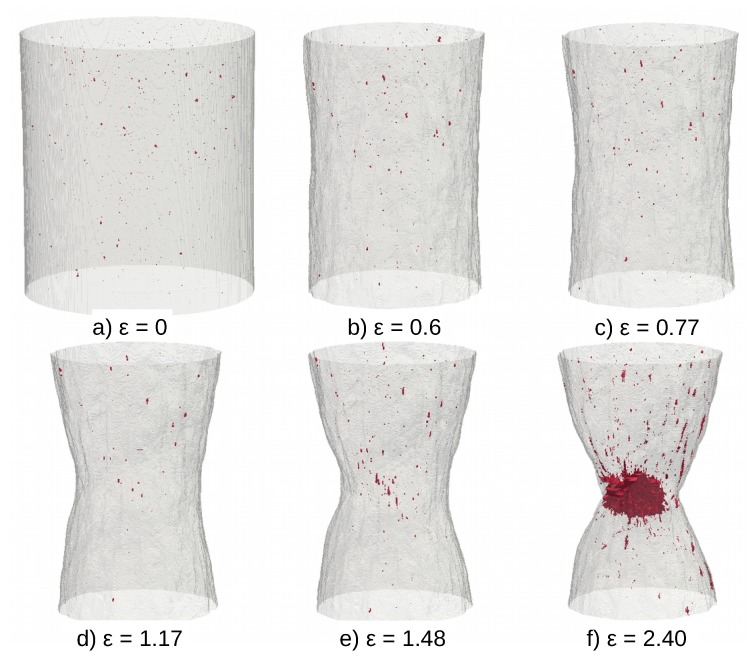
Damage evolution in the non-charged AISI316L sample observed as a volume rendering. The outer surface of the sample is transparent grey and the cavities are the dark red dots. Damage nucleates then grows and finally coalesce in the bottom right image (**a**) initial state; (**b**) true strain ϵ = 0.6; (**c**) ϵ = 0.77; (**d**) ϵ = 1.17; (**e**) ϵ = 1.48; (**f**) ϵ = 2.40.

**Figure 4 materials-12-01426-f004:**
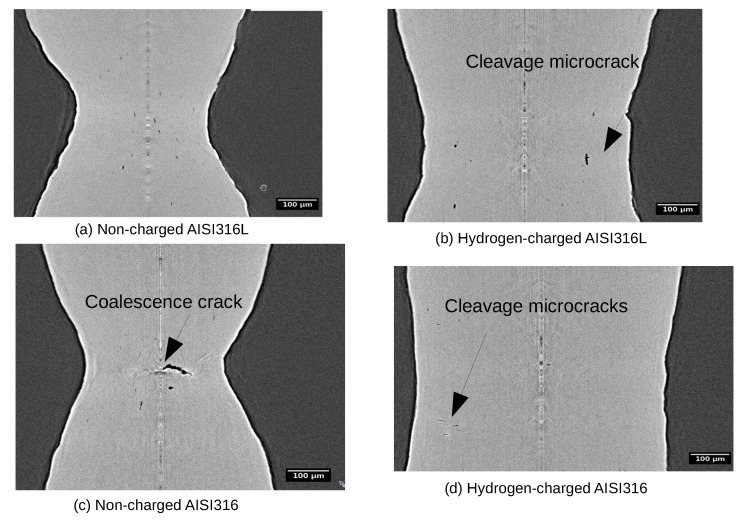
Reconstruction slices extracted parallel to the tensile axis in a central plane for the four samples in the ultimate state before fracture. AISI316 is clearly less deformed at fracture and contains local cleavage microcracks when hydrogen charged. The deformation in the different images are (**a**) ϵ = 2.40 for non-charged AISI316L; (**b**) ϵ = 1.36 for hydrogen-charged AISI316L; (**c**) ϵ = 1.92 for non-charged AISI316; (**d**) ϵ = 0.95 for hydrogen-charged AISI316.

**Figure 5 materials-12-01426-f005:**
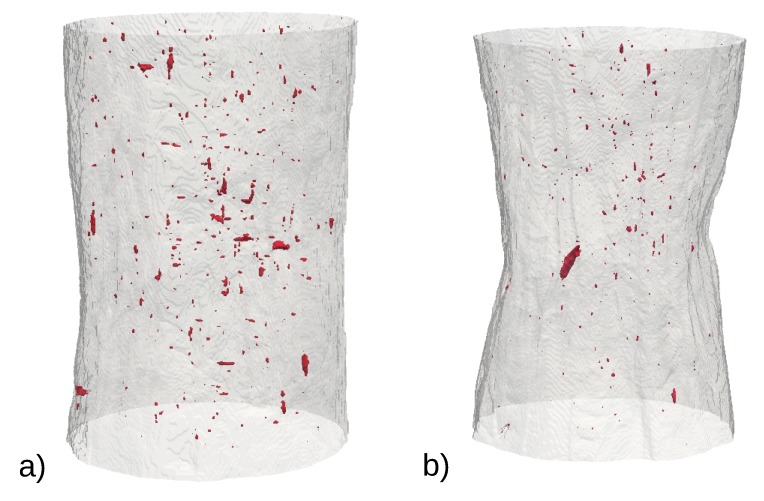
Volume rendering (**a**) hydrogen-charged AISI316 sample just before coalescence. The amount of penny shaped microcracks is very important. The deformation in the image is ϵ = 0.79; (**b**) hydrogen-charged AISI316L sample just before coalescence. The morphology of the cavity is elongated along the tensile direction. The deformation in the image is ϵ = 1.15.

**Figure 6 materials-12-01426-f006:**
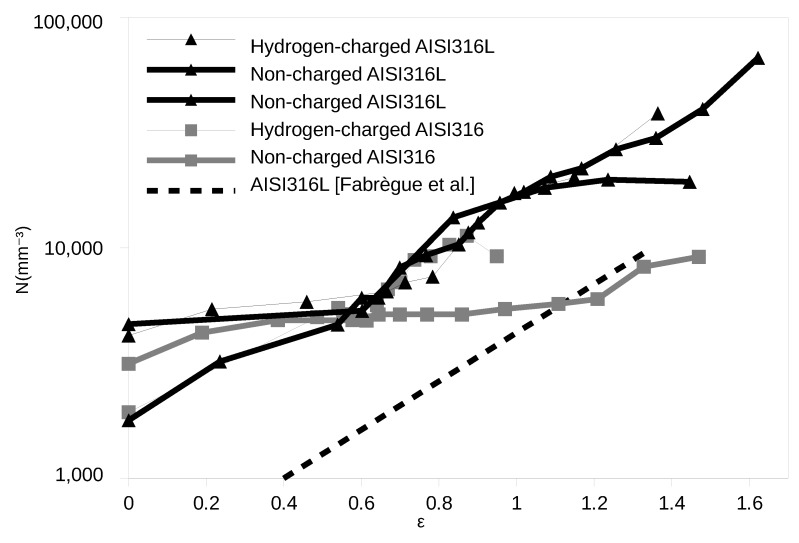
Nucleation quantified as the evolution of the density of cavities with strain. The nucleation in the non-charged AISI316 sample is anomalously small because of a small amount of nucleation in the necked region for this sample. From the behavior of the three other materials, it can be concluded that hydrogen charging has a weak effect on the nucleation kinetics. The measurements by Fabrègue et al. [17] are also shown as a dotted line. These were made using a larger voxel size (1.6 μm pixel size) which explains the lower level of nucleation detected, but the slope of the curve is in line with the measurements of the present paper.

**Figure 7 materials-12-01426-f007:**
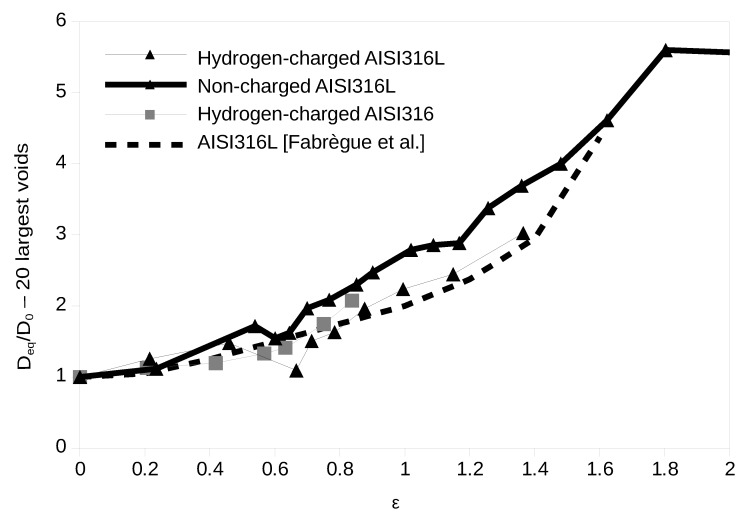
Growth of the average equivalent diameter of the 20 largest cavities with strain.

**Figure 8 materials-12-01426-f008:**
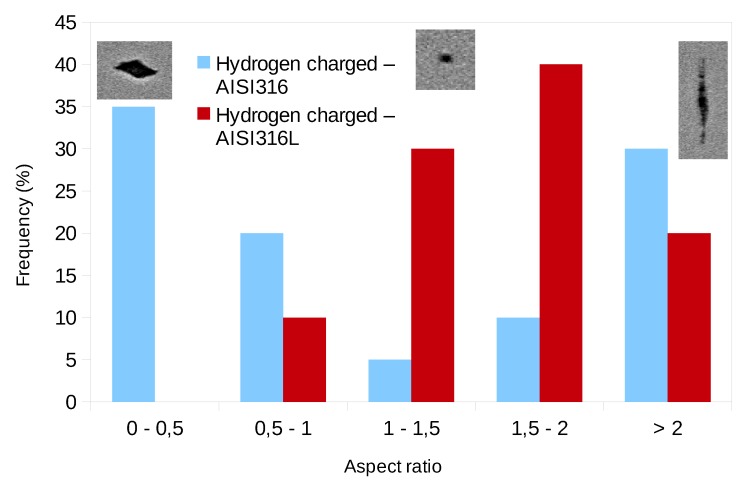
Histogram of the distribution of the aspect ratio of the 20 largest cavities in the two hydrogen charged materials in the state just before fracture.

**Figure 9 materials-12-01426-f009:**
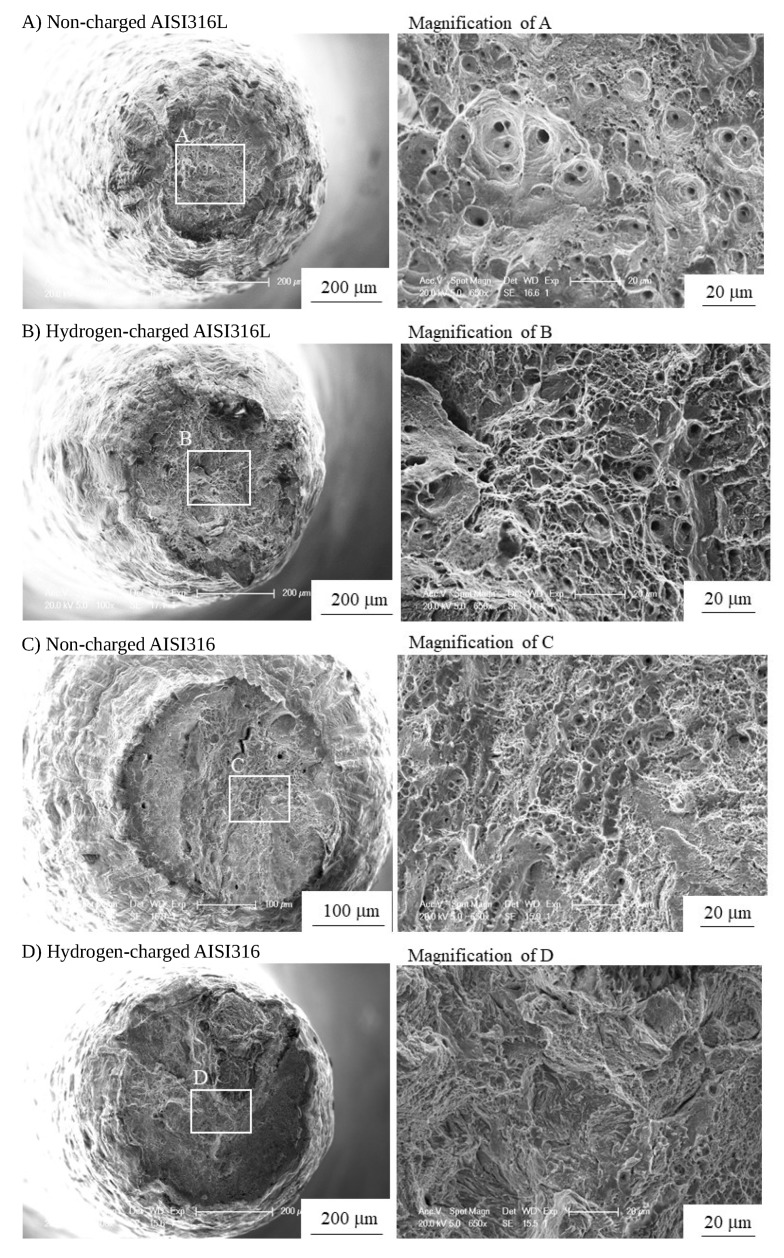
Fractography of non charged and hydrogen-charged samples.

**Figure 10 materials-12-01426-f010:**
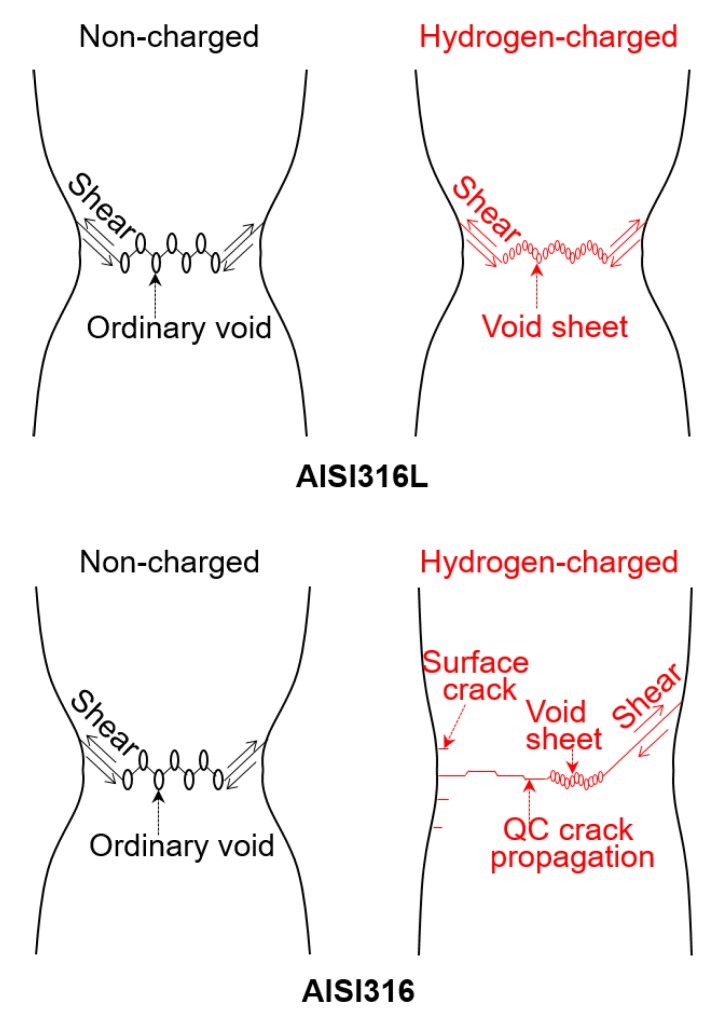
Schematic illustration of fracture mechanism in non-charged and hydrogen-charged AISI316 and 316L.

**Table 1 materials-12-01426-t001:** Chemical composition (mass %) of the two materials.

	C	Si	Mn	P	S	Cr	Mo	Ni
AISI316	0.04	0.64	0.93	0.032	0.001	16.83	2.05	10.23
AISI316L	0.018	0.50	0.84	0.021	0	17.45	2.05	12.09

**Table 2 materials-12-01426-t002:** Tensile properties of the two materials, as provided on the certification sheet by the manufacturer.

Materials	0.2% Proof Stress	Tensile Strength	Elongation (%)
	$\sigma_{0.2}$ (MPa)	$\sigma_{B}$ (MPa)	$\varepsilon_{t}$ (%)
AISI316	263	586	61.0
AISI316L	229	528	66.0

**Table 3 materials-12-01426-t003:** Ductility measured during the in situ tensile tests and calculation of the Decrease in Ductility due to Hydrogen charging (DDH) for the two materials and the two specimen shapes.

Materials	Specimen Shape	Ductility		DDH: Decrease Due to Hydrogen Charging (%)
		Non-Charged	Hydrogen-Charged	
AISI316	Smooth	1.91	0.885	53.7
	Notched	1.66	0.62	62.7
AISI316L	Smooth	1.92	1.35	29.4
	Notched	1.9	1.08	42.9

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
