# Peer review of "Role of Hydrogen-Charging on Nucleation and Growth of Ductile Damage in Austenitic Stainless Steels"

_materials, 2019, doi:10.3390/ma12091426_

Reviewer 1 Report

This paper (Role of Hydrogen-charging on nucleation and growth of ductile damage in Austenitic Stainless Steels) presents interesting results but needs a thorough revision before being considered for publication. Some sections need to be completely rewritten, like the Introduction, literature review and Discussion.

Introduction: The theoretical, analytical and standard approaches should be discussed.

The novelties have to be outlined. It has to be completely rewritten so that the focus of the work and its innovative content can be really appreciated.
Literature review: The Literature review is now a mere list of information but the authors have to provide their own "unifrying" view and not only citing previous work.

In addition, in a quick search I found a number of papers on this topic that you did not cite. I am listing them here, please consider them in the literature review and in the interpretation of the results:                                                                                                                                            

1.       A statistical assessment of ductile damage in 304L stainless steel resolved using X-ray computed tomography, Materials Science and Engineering: A, Volume 728, 13 June 2018, Pages 218-230; A. J. Cooper, O. C. G. Tuck, T. L. Burnett, A. H. Sherry

2.       Local approach to stress relaxation cracking in a AISI 316L-type austenitic stainless steel: Tomography damage quantification and FE simulations, Engineering Fracture Mechanics, Volume 183, 1 October 2017, Pages 170-179, H. Pommier, E. P. Busso, T. F. Morgeneyer, A. Pineau

3.       The effect of phase transformation in the plastic zone on the hydrogen-assisted fatigue crack growth of 301 stainless steel, Materials Characterization, Volume 112, February 2016, Pages 134-141, T. C. Chen, S. T. Chen, W. Kai, L. W. Tsay

4.       Effects of dissolved hydrogen and surface condition on the intergranular stress corrosion cracking initiation and short crack growth behavior of non-sensitized 316 stainless steel in simulated PWR primary water, Corrosion Science, Volume 118, April 2017, Pages 143-157, Xiangyu Zhong, Shirish Chandrakant Bali, Tetsuo Shoji

5.       Effect of austenite instability on the hydrogen-enhanced crack growth of austenitic stainless steels, Corrosion Science, Volume 49, Issue 7, July 2007, Pages 2973-2984, L. W. Tsay, S. C. Yu, R. -T. Huang

6.       The effect of sensitization on the hydrogen-enhanced fatigue crack growth of two austenitic stainless steels, Corrosion Science, Volume 50, Issue 5, May 2008, Pages 1360-1367, L. W. Tsay, Y. -F. Liua, R. -T. Huang, R. -C. Kuo

Results and discussion: The paper presents a few amount of results from unusual experiments but without a theoretical and practical approach.

Conclusions: The discussion about technological benefit have to be separated in the article according points of conclusions. The analysis of the results is quite basic and deserves better and deeper processing.

Author Response

file attached with answers in red

Reviewer 2 Report

The submitted manuscript entitled ‘Role of Hydrogen-charging on nucleation and growth of ductile damage in Austenitic Stainless Steels’ deals with the in-situ tensile test of X-ray irradiated and hydrogen exposed AISI316 and AISI316L austenitic stainless steels. The manuscript sounds and above average, during its careful review only a list of technicalities (detailed below) arose.

 - Please solve every abbreviation at its first occurrence, even if they are well known.

- Please use ‘×’ instead of ‘x’ in geometrical dimensions.

- Please always let a space between the value and its unit, except in the case of ‘°C’ and ‘%’.

- ‘We know from previous experiments that this results in a hydrogen content of 99.7 mass ppm with a uniform distribution over the cross section of the specimen.’ – please provide reference.

- Regarding table 2: (i) how was these values measured (please give reference if the values were taken from different source), (ii) please add scatter to the values.

- What was the real diameter of the tensile samples? They were machined to 1 mm and then further polished.

- The average grain size is reported to be 0.1 mm in section 3.1. Can it have effect on the tensile properties, due to the quite large average grain size compared to the diameter of the sample?

- Please label the subfigures in Figure 1 ((a), (b), …).

- Figure 2 should start from the origin.

- ‘These curves are in rather good agreement with macroscopic tensile test results summarized in Table 2.’ – please compare in tabular form.

- In the opinion of this Reviewer, equation 3 would have more meaning, if it is divided by the non-charged true strain at fracture.

- Please label the subfigures in Figure 3 ((a), (b), …), the subfigure captions (epsilon) are too small to read.

- Please label the subfigures in Figure 4 ((a), (b), …).

 Author Response

file attached with answers in red

Reviewer 3 Report

1.       Fig. 1 – what was the size of representative volume/surface element?

2.       Evade personal forms of English (“our” etc.), as well as “the” in front of “damage” – it is an uncountable noun.

3.       Quality of Figs 6-8 could be better (the lines are enormously thick). Error bars are welcome.

4.       Section 5 could be longer.

Author Response

file attached with answers in red

Reviewer 4 Report

Some minor revisions should be done:

- Lines 42-43: the sentence "Using ... fracture [16]" should be revised in terms of the tenses employed.

- Lines 65-66: "we know form previous experiments ..."; a reference in terature to these experiments should be included.

- Lines 70-72: The ppm of H2 contents after freezing the samples and just before the tests (after some months being frozen) should be given, in order to prove that there was not an impportant diffussion out of the sample. Did you perform H2 content tests after several months frozen.

- Epigraph 2.2: Were the samples machined before H2 charging and freezing? If so, explicit in in the text, please.

- Epigraph 2.2: The samples are 1mm of dimeter but, how is possible that the heads were threaded M3? If the sample has a wider part in the edges, it will be appropiate to include an schematic or plan of the geometry of the samples machined.

- Line 100: Did you do H2 ppm content of the samples after testing? This will be useful in order to know how much was diffused outside during the testing.

- Figure 2: Why the curves do not start in 0,0? Why do they start arround 300MPa , 0.2 true strain?

- Table 3: Could you find an explanation to the value of DDH of smooth 316L that is 29.4% while others are arround 40%-60%?

- Epigraph 5: Could you reccomend what steel 316 or 316L is better to be used for certain applications as FCV vehicules tanks for example? It will be useful.

- Why Figures 9 and 10 are placed at the end of the paper, after the references? The should be placed with the rest of results.

- References 17 and 20: Visitis to websites are included. If possible, it will be more convenient to find research papers about this. 

Author Response

file attached with answers in red

Reviewer 5 Report

Paper deals with important topic of hydrogen embrittlement in steels. Compared are 2 different stainless steels with very nice results and discussion. However, to fully inform reader some additional explanations should be added, and are listed below:

1) In experimental method please give size of tensile specimen used to determine non H charged conditions. Also EBSD experimental setup and sample preparation is not given.

2) Just from interest it would be nice to compare non charged EBSD with charged EBSD, and described changes.

3) Spell check is required. For example, page 7, line 164, page 9, line 213 and page 10 line 239.

2) References, need to be checked for proper name abbreviations. For example, ref 9 states wrong abbreviation. It should give authors as "M. Koyama, T. Ogawa, D. Yan, Y. Matsumoto, C. C. Tasan, K. Takai, K. Tsuzaki". Similar goes for refs 5, 6, 7, 8, 10

Author Response

file attached with answers in red
